# Effect of *Borrelia burgdorferi* Outer Membrane Vesicles on Host Oxidative Stress Response

**DOI:** 10.3390/antibiotics9050275

**Published:** 2020-05-25

**Authors:** Keith Wawrzeniak, Gauri Gaur, Eva Sapi, Alireza G. Senejani

**Affiliations:** Department of Biology and Environmental Science, University of New Haven, West Haven, CT 06516, USA; kwawr1@unh.newhaven.edu (K.W.); ggaur2@unh.newhaven.edu (G.G.); ESapi@newhaven.edu (E.S.)

**Keywords:** outer membrane vesicles (OMVs), reactive oxygen species (ROS), superoxide dismutase 2 (SOD2)

## Abstract

Outer membrane vesicles (OMVs) are spherical bodies containing proteins and nucleic acids that are released by Gram-negative bacteria, including *Borrelia burgdorferi*, the causative agent of Lyme disease. The functional relationship between *B. burgdorferi* OMVs and host neuron homeostasis is not well understood. The objective of this study was to examine how *B. burgdorferi* OMVs impact the host cell environment. First, an in vitro model was established by co-culturing human BE2C neuroblastoma cells with *B. burgdorferi* B31. *B. burgdorferi* was able to invade BE2C cells within 24 h. Despite internalization, BE2C cell viability and levels of apoptosis remained unchanged, but resulted in dramatically increased production of MCP-1 and MCP-2 cytokines. Elevated secretion of MCP-1 has previously been associated with changes in oxidative stress. BE2C cell mitochondrial superoxides were reduced as early as 30 min after exposure to *B. burgdorferi* and OMVs. To rule out whether BE2C cell antioxidant response is the cause of decline in superoxides, superoxide dismutase 2 (SOD2) gene expression was assessed. SOD2 expression was reduced upon exposure to *B. burgdorferi*, suggesting that *B. burgdorferi* might be responsible for superoxide reduction. These results suggest that *B. burgdorferi* modulates cell antioxidant defense and immune system reaction in response to the bacterial infection. In summary, these results show that *B. burgdorferi* OMVs serve to directly counter superoxide production in BE2C neurons, thereby ‘priming’ the host environment to support *B. burgdorferi* colonization.

## 1. Introduction

Lyme disease is the most common vector-borne disease in the United States and Europe [1,2]. The causative agent of the disease, the spirochetal bacterium *Borrelia burgdorferi*, is transmitted through bites from ticks belonging to the genus *Ixodes* [3]. According to the Center for Disease Control, there are 300,000 cases of Lyme disease annually, making it the most common tick-borne illness in the United States [4]. The prevalence and steady growth in Lyme disease cases makes this a pressing public health concern [5]. Lyme disease is a multi-system infection resulting in localized or disseminated acute or chronic symptoms such as erythema migrans, neurological or cardiac involvement, and arthritis [6]. If left untreated, *B. burgdorferi* can colonize the central nervous system (CNS) in a condition known as Lyme neuroborreliosis (LNB) [7,8]. About half of patients with LNB have cranial nerve palsy followed by some experiencing radiculitis, encephalitis, myelitis, and meningitis [9].

Several studies have suggested that *B. burgdorferi* can persist on exposure to unfavorable conditions such as osmotic, pH, and nutrient and temperature changes by changes in morphology and organization, including forming biofilms, cysts, and round bodies [10,11,12,13]. *B. burgdorferi* can also secrete outer membrane vesicles (OMVs), which can help it to interact and alter its surrounding environment [14]. OMVs are constitutively secreted bodies (~50–250 nm in diameter) derived from the outer membrane of both Gram-negative and Gram-positive bacteria [15,16]. They serve as vessels for the transport of DNA fragments, RNA, and proteins [17,18,19]. OMVs can have various functions in the bacterial life style such as to deliver their signaling molecules into a host, to serve as weapons in inter-species environmental competition, and to release misfolded proteins [20], and stressors such as temperature, antibiotics, and nutrient depletion can increase their production [20].

The composition and content of *B. burgdorferi* OMVs have been investigated; however, knowledge of their ex vivo effects on host cells is still limited [18,21]. Several biopsy and microscopic examination studies have shown OMVs are present within human tissues and near host cells, and they have also been observed at sites distant from the bacterial infection site [19]. Past studies show that bacterial OMVs can stimulate secretion of cytokines TNFα, IL-8, and IL-β1 from macrophages [22]. Further, *B. burgdorferi* is known to induce cytokine and chemokine secretion in monocytes and T-cells [23,24]. Innate immunity is known to detect foreign antigen molecules, triggering signaling pathways that result in rapid secretion of myriad inflammatory cytokines and chemokines. However, little is known about the immune response to OMVs and the role they play in disease development.

Neurons secrete several cytokines that are involved in initiating immune responses as well as mediating neuroprotective processes [25,26]. Pro-inflammatory cytokines are known to promote oxidative stress, which is defined as an imbalance between harmful reactive oxygen species (ROS) and the antioxidants that keep their levels tightly regulated [27]. ROS are important in redox biology, protein modification, cell signaling, and pathology [28,29,30]. While cytokine secretion is known to play a role in ROS accumulation, it is unclear whether neuronal cytokine secretion can lead to oxidative stress. Neurons are major producers of ROS, mostly through dynamic mitochondrial activity, which drives the high metabolic rate and ATP generation needed to sustain neuronal functions [31]. However, besides being the site of ATP synthesis, mitochondria are increasingly recognized for their ability to monitor and respond to bacterial infection. They are among the major players in cell signaling and regulation of innate immune responses through ROS production [32]. The capacity of *B. burgdorferi* to alter mitochondrial function is not well understood.

Here, we show that mitochondrial superoxide production in BE2C cells is reduced upon exposure to *B. burgdorferi* spirochetes. Further, reduced superoxide production is observed after exposure to *B. burgdorferi*-derived OMVs alone. On the basis of these results, we propose a model in which *B. burgdorferi* OMVs serve to counter superoxide production in BE2C neurons, thereby ‘priming’ the host environment to make it more hospitable for *B. burgdorferi* colonization and persistence.

## 2. Results

### 2.1. BE2C Cells Internalize B. burgdorferi over Time

To assess the internalization of *B. burgdorferi* by BE2C cells, CM-Dil and anti-*B. burgdorferi* antibodies were used to observe the relative position of *B. burgdorferi* and the BE2C plasma membrane. At 30 min and 2 h co-culture durations, adherence of *B. burgdorferi* to the outer membrane of BE2C cells was observed (Figure 1A,B). At 24 h post infection, *B. burgdorferi* was found to enter the intracellular space of BE2C cells (Figure 1C). To confirm these results, flow cytometry was used to detect green fluorescent protein (GFP)-labeled *B. burgdorferi* that adhered to, or were internalized by, BE2C cells after 24-h co-culture (Figure 1D). Additionally, confocal microscopy and depth coding also confirmed the internalization of *B. burgdorferi* into HEK293 cells after 24 h co-culture (Figure 2). Furthermore, *B. burgdorferi* appeared to form an atypical round body-like morphology (Figure 2).

After confirming *B. burgdorferi* internalization at 24 h, the growth rates of BE2C cells were measured to determine whether *B. burgdorferi* affects cell viability. After co-culturing BE2C cells with *B. burgdorferi* for 30 min, 2 h, and 24 h, BE2C cell viability was evaluated by Alamar blue methods every 24 h over a six-day time period. The results indicated that BE2C cell viability was not changed significantly when untreated cells were compared with cells co-cultured with *B. burgdorferi* (Appendix A).

Apoptosis in BE2C cells co-cultured with *B. burgdorferi* or OMVs for 24 h were assessed via flow cytometry. Cells co-cultured with *B. burgdorferi* for 24 h showed elevated levels of annexin/propidium iodide positive cells compared with the untreated control cells, indicating increased levels of cell death (Appendix A).

### 2.2. BE2C Cell MCP-1 and MCP-2 Secretion Increases after Borrelia burgdorferi Co-Culture

*B. burgdorferi* is known to trigger several cellular responses including the expression of inflammatory molecules, immune mediators, and apoptotic pathways [33,34,35]. To assess host cell response, we assessed cytokine secretion at 24 h post-internalization of *B. burgdorferi*. Release of neuronal cytokines under co-culturing conditions has been reported to alter communication between other neurons and immune cells [25,36]. A commercial 42-plex antibody array was used to assess BE2C cell cytokine/chemokine response after exposure to *B. burgdorferi* for 24 h (Figure 3A,B). The results of quantitative densitometric analysis showed that BE2C cell supernatants had increased levels of three of several chemokines studied in this array (Figure 3C). Specifically, MCP-1, MCP-2, and oncostatin M (OSM) chemokines showed a 232%, 321%, and 39% increase in secretion after *B. burgdorferi* co-culture, respectively.

### 2.3. BE2C Cell Superoxide Levels Decrease after Borrelia burgdorferi and OMV Exposure

Elevated secretion of MCP-1 has previously been associated with increased oxidative stress [37,38]. This prompted an investigation of ROS, namely the mitochondrial superoxide anion radicals, which are known to be one of the most abundant species [39]. BE2C cell superoxide levels were measured after several *B. burgdorferi* co-culture time points using flow cytometry. Increasing co-culture durations with *B. burgdorferi* resulted in significantly decreased superoxide levels at 30 min, 2 h, and 24 h when compared with untreated cells (Mann–Whitney test, *p* < 0.05; Figure 4A). Similar reduction in superoxide was observed in co-cultured HEK293 cells at the same time points [40], even though *B. burgdorferi* internalization was only observed at 24 h (Figure 2A). These results suggest that internalized extracellular factors might be responsible for the rapid superoxide reduction. To see if isolated *B. burgdorferi* OMVs could independently lower superoxide levels, OMVs were isolated using an established ultracentrifugation method [41], and we treated BE2C cells with increasing amounts of OMVs at similar time points. The purity of the *B. burgdorferi* OMV isolates was verified by the presence of the OMV marker OspA and the absence of the intracellular marker DnaK (Figure 4B). After BE2C cell exposure to OMVs, reductions in superoxide levels were observed relative to the untreated control samples (Figure 4C). Interestingly, superoxide levels did not significantly differ as OMV exposure times were increased. BE2C cells were also exposed to differing amounts of OMVs for 30 min to see if a dose-dependent relationship existed (Figure 4D). Again, superoxide levels did not significantly vary with increasing OMV amounts.

### 2.4. BE2C Cell SOD2 Gene Expression Decreases after B. burgdorferi Co-Culture

To determine whether BE2C cell antioxidant enzymes were responsible for the reduction in superoxide levels, SOD2 mRNA transcript levels were assessed using combined reverse transcription quantitative PCR (RT-qPCR). Gene expression analysis of mitochondrial SOD2 transcripts in BE2C cells after a 30 min, 2 h, and 24 h infection was performed (Figure 5). Amplification curve analysis showed that BE2C cell SOD2 expression in infected cells was significantly lower at all infection time-points in comparison with untreated cells (Mann–Whitney test, *p* ≤ 0.05). These results suggest the antioxidant enzyme SOD2 was not responsible for the reduction in superoxide levels.

## 3. Discussion

*Borrelia burgdorferi* uses many strategies for manipulating its host to enhance its own growth and survival [10,11,12,13]. OMVs represent an extracellular means by which it may influence host cell response and homeostasis to enhance infection [42,43]. Here, we report that *B. burgdorferi* is capable of invading BE2C neuronal cells at approximately 24 h, which coincides with an increase in MCP-1 and MCP-2 chemokine secretion. Despite internalization, BE2C cell viability and levels of apoptosis were unchanged. While MCP-1 production has been linked to oxidative stress, we found that mitochondrial superoxide was reduced after co-culture with *B. burgdorferi* and exposure to OMVs. Importantly, this reduction was not because of increased activity of the BE2C cell antioxidant enzyme SOD2, at least as reflected by mRNA levels.

BE2C and HEK293 neuronal cells exposed to *B. burgdorferi* for 24 h showed intracellular localization of *B. burgdorferi* (Figure 2 and Figure 3), which is in agreement with past research showing internalization in neuroglial cells, fibroblasts, and macrophage cell lines at similar time points [44,45]. Cell viability and apoptosis were not impacted after six days of co-culture, which has been similarly shown in other studies [46]. In addition, *B. burgdorferi* at 24 h time points appeared to take on an atypical, round body-like morphology (Figure 2). Altered *B. burgdorferi* morphology has been noted to take place on heating, such as co-culturing at 37 °C, 5% CO_2_, and the application of other chemical stressors [10]. Altered morphology could also reflect the nutrient poor intracellular space. Round body-like morphologies, which may aid *B. burgdorferi* in its survival, have been described in fibroblasts after co-culture [44]. However, the intracellular space alone may offer protection as intracellular *B. burgdorferi* are found to be more viable after treatment with gentamycin [44]. Importantly, for this study, the intracellular localization of *B. burgdorferi* at ~24 h in mammalian cells is informative in that it suggests that observed effects could be because of intracellular interactions.

Alteration of reactive oxygen species can initiate proinflammatory cytokine production, leading to an immune response. An antibody array was used to probe several cytokines and chemokines associated with the immune response and oxidative stress in human cells. MCP-1 and MCP-2 chemokines were shown to be markedly increased in BE2C cell co-culture supernatant after 24 h (Figure 4). MCP-1 is implicated in normal brain physiology, as well as in response to oxidative stress [47,48]. It is also known to be elevated to promote the infiltration of monocytes, T lymphocytes, and natural killer cells into areas of infection [48]. Dorsal root ganglia cells also increase MCP-1 secretion upon *B. burgdorferi* exposure [8]. Another study showed that oncostatin M (OSM), a cytokine that has been associated with cancer proliferation, neuronal development, and CNS inflammatory diseases, was also elevated after *B. burgdorferi* exposure [49]. Additional studies have documented the secretion of cytokines when cells are exposed to *B. burgdorferi.* For example, dorsal root ganglia explants and human Schwann cells exposed to *B. burgdorferi* for 48 h have shown increased levels of IL-6, IL-8, and CCL2 (MCP-1) [8]. On the basis of the data in this study, MCP-1 and MCP-2 secretion is likely one of the first responses to infection by *B. burgdorferi*.

ROS, such as mitochondrial superoxide, are important in redox biology, protein modification, cell signaling, and pathology [28,29,30].

An imbalance of ROS and antioxidants produces oxidative stress. MCP-1 is well known to attract resident immune cells to the sites of infection and has been shown to be expressed in response to oxidative stress in epithelial cells treated with hydrogen peroxide, another ROS [50,51]. Co-culturing *B. burgdorferi* and BE2C cells resulted in significant reduction in superoxide (Figure 5), suggesting that oxidative stress was not occurring. A reduction in superoxide levels was also observed in BE2C cells as early as 30 min after introduction of OMVs (Figure 5). Superoxide reduction caused by OMV exposure was not as significant as those caused by *B. burgdorferi*. This may be explained by other secreted factors not accounted for or the constitutive secretion of OMVs. In addition, the proximity of *B. burgdorferi* to host cells may encourage a higher proportion of OMVs to interact with host cells compared with passive interactions of OMVs alone in co-culture.

*B. burgdorferi* adhered to BE2C cells as soon as 30 min after co-culturing (Figure 1), which may explain the observed simultaneous reduction in superoxide (Figure 5). In addition, increasing OMV amounts did not show a proportional decline in BE2C superoxide levels; however, reductions took place under all OMV conditions compared with untreated controls. Interestingly, *B. burgdorferi* was shown to enter BE2C cells at 24 h, which is comparable to the time point of internalization in another report [45]. Superoxide reduction was observed as early as 30 min, which is prior to *B. burgdorferi* internalization. This indicates that extracellular OMVs could mediate superoxide reduction in BE2C cells prior to internalization.

Mammalian cells have a robust antioxidant system that has the capacity to lower ROS, including mitochondrial superoxide [51]. SOD2 localizes to the mitochondria, where it converts superoxide to H_2_O_2_ [52]. To rule out the possibility that BE2C cells did not use endogenous antioxidant enzymes, SOD2 gene expression levels were examined at 30 min, 2 h, and 24 h co-culture timepoints (Figure 5). SOD2 mRNA levels were slightly lower in BE2C cells co-cultured with *B. burgdorferi* compared with untreated cells. Despite a decrease in SOD2, superoxide was also decreased in this study (Figure 5), suggesting that BE2C cell antioxidant enzymes were not responsible for the reduction in superoxide.

We can interpret this coincidental decrease in superoxide and SOD2 in two ways: the presence of *B. burgdorferi* superoxide dismutase (SodA), which is found in *B. burgdorferi* OMVs, may serve to neutralize superoxide, which may otherwise activate BE2C SOD2 expression [18]. Many bacteria use such enzymes to safeguard against stressful environmental conditions and immune cell defenses. One example is *Helicobacter pylori* OMVs, which contain a catalase-like protein (KatB) and are shown to be protective against ROS from immune cell respiratory bursts [53]. *B. burgdorferi* uses a superoxide dismutase, which has been shown to be an essential virulence factor for infection of mice [54]. This enzyme is present in *B. burgdorferi* OMVs, making it a prime candidate for modulation of host cell superoxide levels [18]. Past studies have shown that *B. burgdorferi* can reduce peripheral blood mononuclear cell ROS levels in response to ROS-generating stimuli, while at the same time showing a decrease in intracellular superoxide [55]. In the same study, it was found that the decrease in superoxide was independent of mammalian antioxidants, namely glutathione. However, other research looking at mitochondrial superoxide has shown an increase in peripheral blood mononuclear cells (PBMCs) derived from Lyme disease patients [56]. The differences in superoxide levels in this study and in the literature may be because of the use of different cell types. PBMCs, including immune cells, are likely primed by *B. burgdorferi* to produce superoxide to clear the pathogen. The nature of superoxide in neuron-like cell lines in response to culture with bacteria is less clear. However, neurons are large generators of ROS, and thus may be inhospitable for *B. burgdorferi* once internalized [57]. This may explain why prior to internalization, BE2C cell superoxide levels are already in decline. OMVs may prime BE2C cells to reduce superoxide that would otherwise be a source of oxidative stress to *B. burgdorferi*. A second possible interpretation considers the relationship between mRNA transcripts and their translated proteins. However, if indeed SOD2 expression is lowered, this may leave host cells susceptible to ROS generated by immune cells in response to *B. burgdorferi*. This may explain some models showing increased levels of apoptosis when culturing neurons, *B. burgdorferi*, and microglia, but not when co-culturing neurons and *B. burgdorferi* alone [58]. Elevated apoptosis has also been shown owing to inflammation in models using tissue explants and *B. burgdorferi* [8]. When factoring in inflammation and oxidative burst by immune cells, lower antioxidant expression in host neurons caused by *B. burgdorferi* could leave cells without protection and less viable. In this study, BE2C cell viability and apoptosis were not significantly impacted (Appendix A). However, in an in vivo situation, the presence of immune cells may have a greater impact on BE2C cell viability.

In conclusion, *B. burgdorferi* and OMVs cause a reduction of BE2C cell mitochondrial superoxide; this reduction is independent of SOD2 expression in BE2C cells. Internalization data show that superoxide levels decrease prior to entry by *B. burgdorferi*, suggesting that *B. burgdorferi* may be priming BE2C cells prior to entry. The reduction of SOD2 may make neurons even more susceptible to oxidative stresses during immune responses to infection. Furthermore, the ROS diminishing capacity of OMVs may shield *B. burgdorferi* from immune cell ROS burst. In addition, OMVs may divert the immune cells away from the bacteria, leading to a mistargeted and ineffective response. Future research must examine more apoptotic markers and the induction of cytokines and chemokines after OMV exposure over greater co-culture durations. The effects of Borrelia biofilms on host cell response should also be explored. This study highlights the role of superoxide during in vitro infection and the potent role that *B. burgdorferi* OMVs may play in modulating host cell functions and the resulting contribution to pathogenesis.

## 4. Methods

### 4.1. Cell Culture

The neuroblastoma cell line, BE2C (ATCC #CRL-2268, Manassas, VA, USA), was grown in Eagle’s minimum essential medium (EMEM) (Sigma, Cat. no. M0268, St. Louis, MO, USA) and Ham’s F12 (Thermo Fisher Scientific, Cat. no. 21700018, Waltham, MA, USA) supplemented with 10% fetal bovine serum (FBS) and 1% penicillin-streptomycin-glutamine (PSG) at 37 °C, 5% CO_2_ for two days prior to all treatments to allow for cell recovery. HEK293 cells (ATCC #CRL-1573) were grown in Dulbecco’s minimum essential medium (DMEM) (Sigma, Cat. no. D7777) with 10% fetal bovine serum (FBS) and 1% PSG at 37 °C, 5% CO_2_ for two days prior to all treatments to allow for cell recovery. All *B. burgdorferi* infections were carried out in a co-culture media composed of EMEM/F12 or DMEM, both supplemented with 6% rabbit serum. BE2C and HEK293 cells beyond passage number 20 were not used in experiments.

*B. burgdorferi* B31 (ATCC) was grown in Barbour–Stoenner–Kelly (BSK-H) media supplemented with 6% rabbit serum (Pel-Freez Biologicals, Rogers, AR) at 33 °C, 3% CO_2_. *B. burgdorferi* beyond passage number 6 (P6) were not used in any of the experiments. A Petroff–Hauser counter was used to count bacteria for infection studies. To collect *B. burgdorferi* bacteria for experiments, cultures were centrifuged at 300× *g* for 10 min at room temperature (RT) in a microcentrifuge and cells were resuspended in fresh BSK-H, 6% rabbit serum. All infections were carried out at a multiplicity of infection (MOI) of 40 with 30 min, 2 h, and 24 h time points, based on previously published methods [8,21].

### 4.2. Isolation of Outer Membrane Vesicles (OMVs)

OMV isolations were performed by adapting methods previously described [40]. *B. burgdorferi* were grown in autoclaved 10 mL glass culture tubes to a total volume of 200 mL for seven days. OMVs were isolated from a total of 10^9^–10^10^ bacteria. Cultures were pooled and centrifuged at 3200 × *g* for 60 min in a Beckman GS-6R centrifuge. The supernatant was removed, and the pellet was resuspended in fresh BSK-H with 6% rabbit serum. Bacteria were then incubated at 37 °C, 5% CO_2_ for 2 h to promote OMV production. Cultures were pooled again, and the bacteria were pelleted by centrifugation at 3200× *g* for 60 min. The supernatant was removed and double filtered using a 0.22 µm polyethersulfone (PES) filter (Millipore Cat no: SLGP033RS, Burlington, MA, USA). The bacterial pellet was resuspended in lysis buffer (50 mM Tris–HCl (pH 7.4), 150 mM NaCl, 1 mM EDTA, 1% SDS), the insoluble portion was pelleted, and the remaining supernatant was stored at −80 °C. For ultracentrifugation, polycarbonate tubes (Beckman #355618, Brea, CA, USA) were used with a type 70 Ti rotor (Beckman #337922, Brea, CA, USA). The supernatant was ultracentrifuged using a Beckman Optima LE-80k ultracentrifuge for 1 h at 100,000× *g*, 4 °C. The pellet was resuspended in PBS, aliquoted, and stored at −80 °C until needed.

### 4.3. Immunofluorescence Microscopy

Using 35 mm polystyrene culture dishes, 1 × 10^4^ BE2C or HEK293 cells were grown on autoclaved coverslips for two days. The medium was then replaced with 4 × 10^5^
*B. burgdorferi* cells suspended in 2 mL of EMEM/F12 supplemented with 6% rabbit serum. Cells were co-cultured at 37 °C, 5% CO_2_ for various times (30 min, 2 h, and 24 h). At each time point, co-culture media were removed and cells were washed twice with PBS. A 10 µg/mL solution of Cell Tracker CM-Dil (Thermo Fisher Sci., Cat. no. C7001, Waltham, MA, USA) in PBS was added to the culture dishes. Cells were incubated at 37 °C for 5 min, and then transferred to 4 °C for 15 min. The dye solution was then aspirated, and the cells were washed once in PBS. Cells were then fixed using ice-cold methanol for 5 min at −20 °C. Methanol was aspirated and cells were washed once in PBS. FITC conjugated anti-*Borrelia burgdorferi* polyclonal antibodies (Thermo Fisher Sci, Cat. no. PA1-73005, Waltham, MA, USA) were applied at a 1:200 dilution in PBS and cells were incubated at 37 °C for 30 min. Cells were then washed with PBS and DAPI was applied at a 1:1000 dilution in PBS for 5 min at RT. Cells were then washed in PBS and the slides were mounted with Permaflour mounting medium (Thermo Fisher, TA-030-FM, Waltham, MA, USA). A Nikon Eclipse 80i microscope (Nikon, Tokyo, Japan) with NIS-Elements software, as well as Leica SP8 (Leica, Wetzlar, Germany) with LAS X software, were used for imaging and analysis.

### 4.4. Cytokine Array

BE2C cells were plated at a density of 1 × 10^5^/mL and grown for two days in 60 mm tissue culture dishes. The medium was then replaced with *B. burgdorferi* in serum-free EMEM/F12 for 24 h at an MOI of 40. An untreated dish of BE2C cells received only fresh serum-free co-culture medium. After 24 h, the co-culture supernatant was centrifuged for 15 min to remove BE2C cells, *B. burgdorferi*, and debris in solution. The supernatant was stored at −80 °C. The supernatant was used for cytokine analysis using the Human Cytokine Array C5 (RayBioTech, cat#: AAH-CYT-5-2, Atlanta, GA, USA) according to the manufacturer’s instructions. ImageJ software was used for all densitometric analyses. Positive control spots on each array were used to normalize the signal of each detected cytokine.

### 4.5. Flow Cytometry

A total of 5 × 10^4^ BE2C cells were seeded per well in 12-well polystyrene plates. For several infection time points, well medium was replaced with 2 × 10^6^
*B. burgdorferi* or 7 µg, 14 µg, or 28 µg of OMVs, as determined by the total protein content, in a volume of 2 mL. *B. burgdorferi* and OMV exposures were carried out for 30 min, 2 h, and 24 h. For OMV exposures, OMV protein content was related to the initial culture concentration to represent ~2 × 10^6^ bacteria.

After all treatments, untreated wells received co-culture medium only. As a positive control, menadione at a final concentration of 1 mM in 2 mL of EMEM/F12, 6% rabbit serum was used. At 30 min prior to cell harvesting, 2 µL of 5 mM Mitosox (Thermo Fisher Sci, M36008, Waltham, MA, USA) in DMSO was added to each well to a final concentration of 5 µM. One well of untreated cells without Mitosox was used to measure background fluorescence. As a vehicle control, 2 µL of DMSO was added to the control wells. Cells were washed in PBS, trypsinized, and centrifuged at 300× *g* for 5 min. The supernatant was removed, and the cells were resuspended in 500 µL of cold flow buffer consisting of PBS, 5 mM EDTA, and 1% FBS. Tubes were kept on ice prior to filtration of cells into sterile 5 mL polystyrene tubes with a cell-strainer cap. Mean fluorescence intensity was used to quantify fluorescence to measure superoxide levels.

Apoptosis studies were performed using an FITC-Annexin V Apoptosis Detection Kit (BD Biosciences, Cat. no. 556547, Franklin Lakes, NJ, USA) according to the manufacturer’s instructions. Menadione was used to induce apoptosis and serve as a positive control. Samples were split into three groups: FITC/PI, FITC, and PI. Untreated unstained samples were used as a negative control. All flow cytometry was performed using a BD Accuri C6 flow cytometer and associated software.

### 4.6. Western Blotting

Total protein content of the OMV isolate was measured using the DC protein assay (Bio-Rad Cat. no. 5000112, Hercules, CA, USA), according to the manufacturer’s recommendations. SDS-PAGE was performed by loading wells with 10 µg of *B. burgdorferi* cell lysate and OMV isolates. Precast 4%–15% Mini-Protean TGX gels (Bio-Rad, Cat. no. 456-1083, Hercules, CA, USA) were run at 100 V for 1.5 h. Gels were transferred to Trans-Blot Turbo mini PVDF (Bio-Rad, Cat. no. 1704156, Hercules, CA, USA) membranes using the Trans-Blot Turbo transfer system. After transfer, the membrane was washed three times in Tris buffered saline with Tween 20 (TBST), then incubated in 5% non-fat dry milk (Bio-Rad, Cat. no. 1706404, Hercules, CA, USA) in TBST for 1 h at RT. Membranes were rinsed once in TBST, then washed three times in TBST in 10 min intervals. Mouse monoclonal DnaK antibodies (CB312) were donated by Dr. Jorge Benach of Stony Brook University and used at a dilution of 1:2 in 5% BSA in TBST. Mouse monoclonal OspA antibody (H-5332, generously provided by Dr. Thomas G. Schwan, Rocky Mountain Laboratories, Hamilton, MT, USA) was used at a dilution of 1:200 in 5% BSA in TBST. The membrane was cut along the 50 kDa line and each half was incubated in their respective primary antibody solutions overnight at 4 °C in a rotating tube incubator. Membranes were rinsed once in TBST, and then washed three times in TBST for 10 min each time. Membranes were incubated in anti-mouse HRP-conjugated secondary antibodies at a dilution of 1:5000. The membrane was incubated for 1 h on a rocker. Membranes were washed in TBST three times in 10 min intervals. Signal enhancement was performed by applying 3 mL of ECL solution (G-Biosciences, Cat. no. 786-002, St. Louis, MO, USA) to the membrane and visualizing on a Fluorochem E imager (ProteinSimple, San Jose, CA, USA).

### 4.7. RT-PCR and qPCR

RNA was extracted from BE2C cells infected with *B. burgdorferi* using Trizol reagent (Thermo Fisher Sci., Cat. no. 15596026) according to the manufacturer’s recommendations. RNA was stored at −80 °C until needed. cDNA synthesis was performed with the Verso cDNA Synthesis Kit (Thermo Fisher Sci., cat#: AB1453A) using anchored oligo-dT primers. The cycling program was as follows: 42 °C for 30 min and 95 °C for 2 min. cDNA was stored at 4 °C for no longer than one week, and then transferred to −20 °C. qPCR was carried out using All-in-One qPCR Mix (GenecoPoeia, Cat. no. QP004, Rockville, MD, USA) in white 96-well plates according to the manufacturer’s recommendations. Primer sequences for SOD2 [59] were made by Integrated DNA Technologies (Coralville, IA, USA) as follows: Forward- 5′ GACAAACCTCAGCCCTAACG, Reverse- 5′ GAAACCAAGCCAACCCCAAC. Primer sequences for internal control GAPDH were as follows: Forward- 5′ GCATCTTCTTTTGCGTCG, Reverse- 5′ TGTAAACCATGTAGTTGAGGT. cDNA equal to 100 ng of initial total RNA was used as a template. Using a Bio-Rad CFX96 (Hercules, CA, USA), the following cycler conditions were used: 10 min at 95 °C once, followed by 15 s at 95 °C, 20 s at 60 °C, and 20 s at 72 °C for 40 cycles. Gene expression analysis was performed using Bio-Rad CFX manager version 3.1.

### 4.8. Statistical Analysis

GraphPad Prism 6 software (San Diego, CA, USA) was used for statistical analysis. Statistical significance was determined using the non-parametric Mann–Whitney test.

## Figures and Tables

**Figure 1 antibiotics-09-00275-f001:**
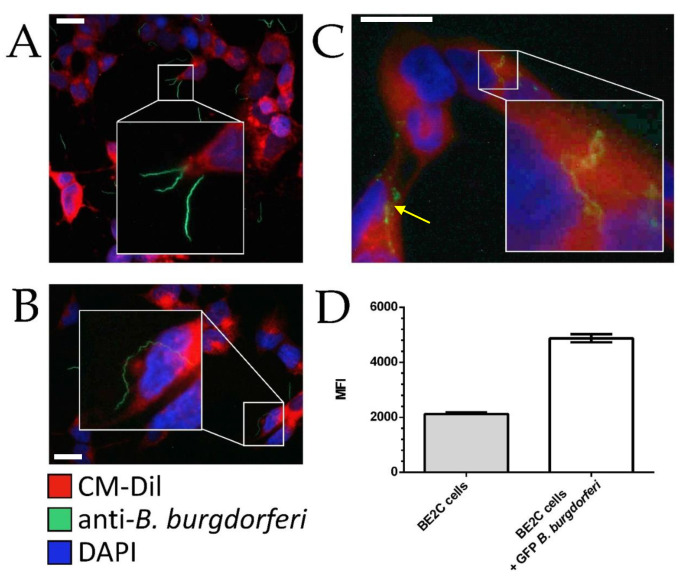
Spatiotemporal interactions between BE2C cells and *B. burgdorferi*. BE2C cell co-culturing at 30 min (**A**), 2 h (**B**), and 24 h (**C**) timepoints were visualized via Figure 2. C cell boundaries while polyclonal anti-*B. burgdorferi* was used to visualize *B. burgdorferi*. (**D**) After 24 h co-culture with green fluorescent protein (GFP)-labeled *B. burgdorferi*, BE2C cells were analyzed by flow cytometry to assess GFP fluorescence, indicating cell–cell interactions. All images were taken at 400× magnification. Scale bar is 25 µm. MFI, mean fluorescence intensity. DAPI, 4’,6-diamidino-2-phenylindole.

**Figure 2 antibiotics-09-00275-f002:**
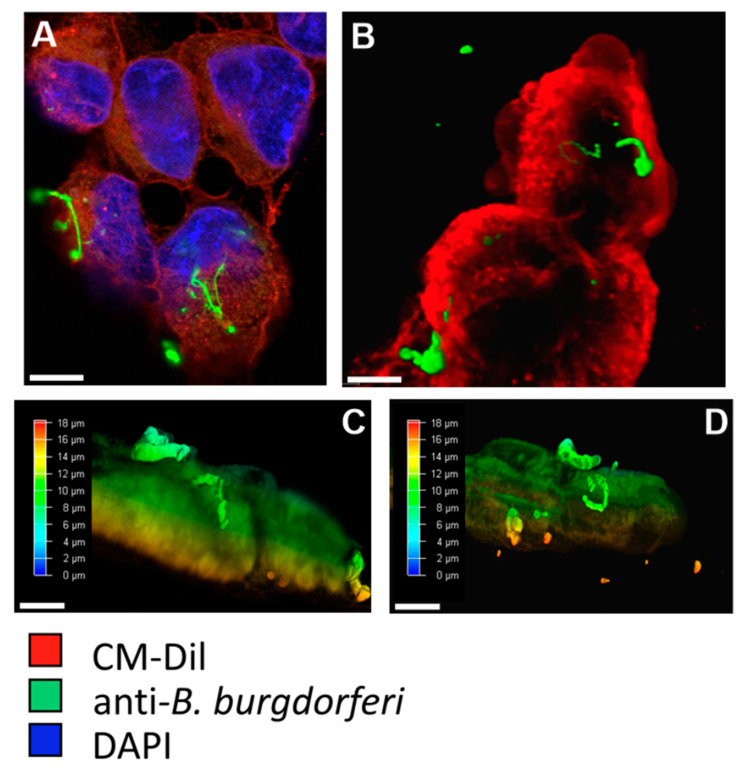
Twenty-four hour *B. burgdorferi* co-culturing with HEK293 cells. Co-cultured HEK293 and *B. burgdorferi* were imaged using confocal microscopy. *B. burgdorferi* are observed to adhere to the outside of HEK293 cells at 24 h (**A**), while 3D imaging revealed that *B. burgdorferi* were internalized by HEK293 cells at 24 h (**B**). Scale bar is 15 µm. Depth coding showed the relative position of HEK293 cells to *B. burgdorferi* (**C**,**D**). All images taken at 1000× magnification. Scale bar is 5 µm.

**Figure 3 antibiotics-09-00275-f003:**
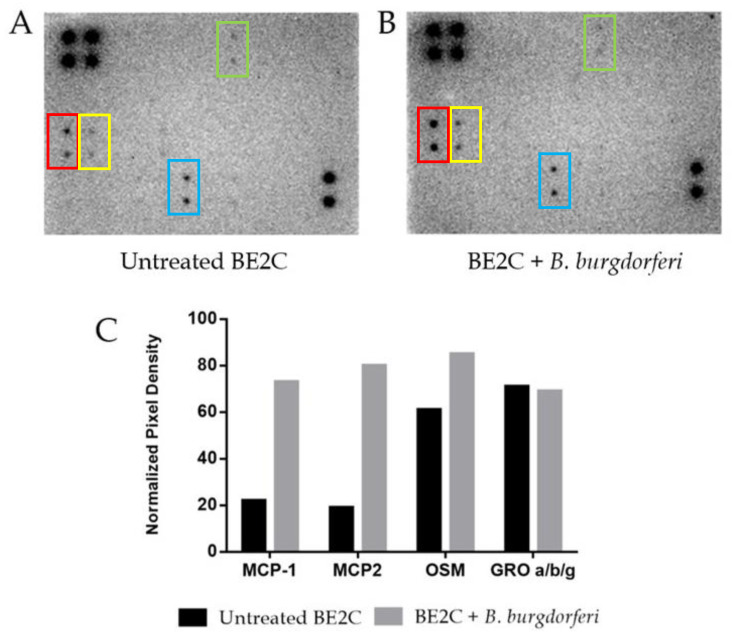
Cytokine secretion by BE2C cells after 24 h co-culture with *B. burgdorferi*. BE2C cells either received fresh co-culture medium (**A**) or *B. burgdorferi* resuspended in co-culture medium (**B**) for 24 h A 42-plex antibody panel detected monocyte chemoattractant protein 1 MCP-1 (red box), MCP-2 (yellow box), oncostatin M (OSM) (blue box), and chemokine GRO a/b/g (green box) in both conditions. (**C**) Densitometric analysis was used to quantify signal intensities between untreated and *B. burgdorferi*-treated cells.

**Figure 4 antibiotics-09-00275-f004:**
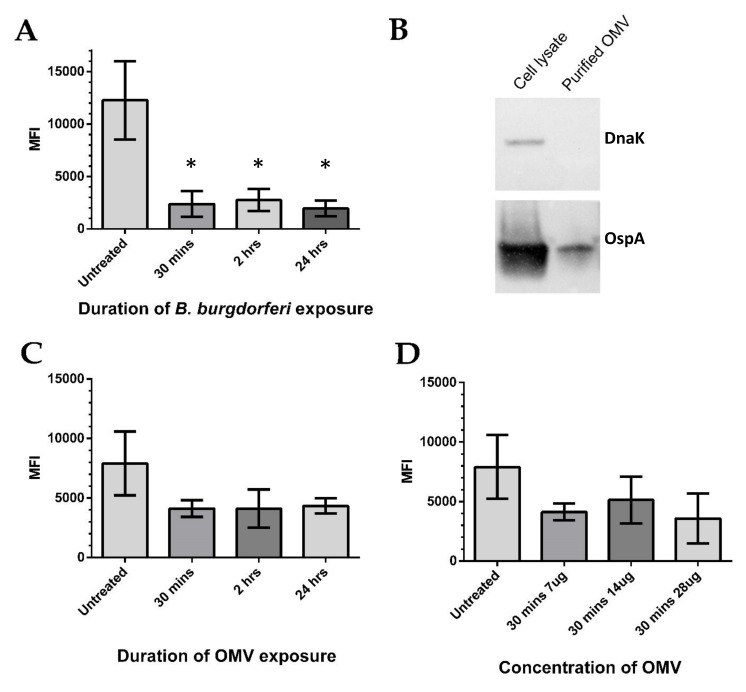
*B. burgdorferi* and outer membrane vesicles (OMVs) decrease BE2C cell mitochondrial superoxide levels. (**A**) Flow cytometry was used to measure the mean fluorescence intensity (MFI) of the mitochondrial superoxide indicator, Mitosox, in BE2C cells. Increasing co-culture times with *B. burgdorferi* results in BE2C cell superoxide reduction. (**B**) *B. burgdorferi* OMVs were isolated via ultracentrifugation and purity verified by observing an absence of the intracellular protein DnaK from OMV isolates and presence of membrane-specific OspA. (**C**) BE2C cell superoxide decreased after treatment with OMVs, but was not statistically significantly compared with the untreated conditions. Increasing OMV exposure durations did not produce a proportional superoxide decrease. (**D**) BE2C cell superoxide levels decreased in comparison with untreated cells, but were not statistically significant compared with the untreated conditions in response to *B. burgdorferi* OMVs at varying OMV amounts. An unpaired Mann–Whitney test was used to determine significance between treated and untreated conditions (* significant at *p* < 0.05). Error bars represent SEM of six experiments.

**Figure 5 antibiotics-09-00275-f005:**
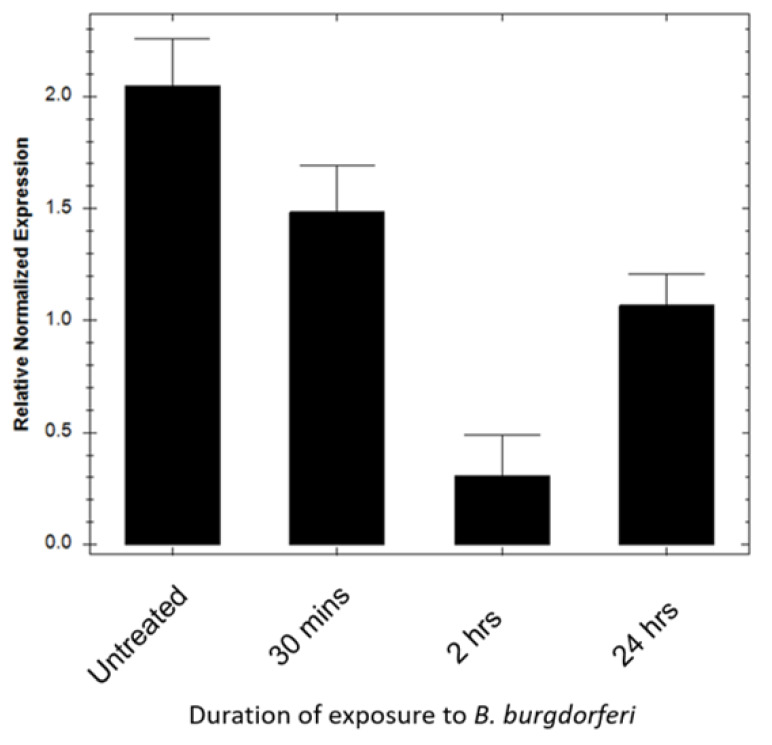
Superoxide dismutase 2 (SOD2) expression in BE2C cells decreases after exposure to *B. burgdorferi*. Reverse transcription quantitative PCR (RT-qPCR) was used to measure gene expression of SOD2 in BE2C cells using human SOD2 specific primers. BE2C cells were co-cultured with *B. burgdorferi* for 30 min, 2 h, and 24 h durations followed by the use of RT-qPCR to establish human specific SOD2 expression. Analysis was performed using Bio-Rad CFX Manager. Error bars represent SEM of three experiments.

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
