# Peer review of "Effect of Borrelia burgdorferi Outer Membrane Vesicles on Host Oxidative Stress Response"

_antibiotics, 2020, doi:10.3390/antibiotics9050275_

Round 1

Reviewer 1 Report

Wawrzeniak et. al. present an interesting work determining the effect of  Outer membrane vesicles (OMVs) in cellular responses against  Borrelia burgdorferi. The authors use the BE2C neuroblastoma cell to evaluate the spirochete internalization and the effect of infection on cytokine production. They also claim that the OMV reduces mitochondrial superoxides production. However, I suggest a more exhaustive description of the statistical test performed in ever experiment to determine significance.

Other comments are found below:

The introduction can benefit from some work. Paragraph structure looks unorganized, for instance:

Line 37: Lyme disease is a multi-system infection, resulting in localized or disseminated acute or chronic symptoms? Such as? Ranging from? Please, specify. Then, in line 43 authors state that the symptoms occur depend on the ability of the Borrelia to evade host responses to eliminate the pathogen. What are the specific symptoms and how are they associated with specific evasion mechanisms? Also, there are no references at the end of this affirmation

Line 48 Persist under stressful conditions, such as?

Line 52: OMV production occurs at all stages of growth? Can the authors please be more specific?

Line55: “The composition and content of OMV have been investigated” Authors then go to comment on the lack of in vitro studies addressing the effect of such composition on the host without mentioning what components are or whether there are in vivo studies reported in the literature.

Line 66: The sentence in short reads “while immune responses are recognized to mediate the reaction to foreign antigens much less is known about the ability of OMV to alter homeostasis in neurons”. This needs to be rewritten in a way that shows a logical flow of information between cell responses, neurons, and OMV.

These are just some examples, but such vague statements are found all through the introduction.

Methods:

Line 312: Please specify that it is Borrelia burgdorferi the pathogen used in the infections in here too or move/condense with line 322.

Line 319: B. burgdorferi can be mentioned once in this sentence

Line 329: Infection timepoints should be described in the method section too.

Line 381: Ultracentrifugation should be renamed "Isolation of Outer membrane vesicles (OMVs)" or something similar and moved after the description of the cell culture methods.

Results:

It is not clear why internalization in  HEK293 cells was made. Especially, when/if part of the reason using this cell line here was to compare reductions in superoxide with the neuroblastoma cells and the data is not shown. I suggest including this data as a comparison if an important claim of this study is that “internalized extracellular factors might be responsible for the rapid superoxide reduction”

Figure 4: unpaired Mann-Whitney test (compare two groups) was used to determine significance between the untreated cells and each time point? This is important because the description of the statistical analysis only mentioned the Kruskal-Wallis test (compare 3 or more groups). I suggest describing in the text (not only in the figure but also in the result section) each test performed to make the reader understand more easily the significance of every comparison. For instance:

“Amplification curve analysis showed that BE2C cell SOD2 expression for infected cells was significantly lower at all infection time-points in comparison to untreated cells (XXX test, p=0.005)”

Line 122: “Quantitative densitometry analysis results showed that BE2C cell supernatants had increased levels of three of several chemokines

Author Response

Author's Reply to the Review Report (Reviewer 1)

We thank the Reviewer for constructive comments on our manuscript. We have addressed each of the concerns in the manuscript as they are explained below.

Wawrzeniak et. al. present an interesting work determining the effect of Outer membrane vesicles (OMVs) in cellular responses against Borrelia burgdorferi. The authors use the BE2C neuroblastoma cell to evaluate the spirochete internalization and the effect of infection on cytokine production. They also claim that the OMV reduces mitochondrial superoxides production. However, I suggest a more exhaustive description of the statistical test performed in ever experiment to determine significance.

We have respectively responded to reviewer point and added more information where statistical method used for each section. Further clarification can be found on page 3 bellow.

Other comments are found below:

The introduction can benefit from some work. Paragraph structure looks unorganized, for instance:

Line 37: Lyme disease is a multi-system infection, resulting in localized or disseminated acute or chronic symptoms? Such as? Ranging from? Please, specify. Then, in line 43 authors state that the symptoms occur depend on the ability of the Borrelia to evade host responses to eliminate the pathogen. What are the specific symptoms and how are they associated with specific evasion mechanisms? Also, there are no references at the end of this affirmation

We addressed this issue by modifying the sentence to: “Lyme disease is a multi-system infection resulting in localized or disseminated, acute or chronic symptoms such as erythema migrans, neurological or cardiac involvement and arthritis [6].”

We also added the following information to address some specific symptoms “About half of patients with LNB have cranial nerve palsy followed by some experiencing radiculitis, encephalitis, myelitis and meningitis [9]. “

Line 48 Persist under stressful conditions, such as?

We added the following in that sentence “…under exposure to unfavorable conditions as osmotic, pH, nutrients and temperature changes….”

Line 52: OMV production occurs at all stages of growth? Can the authors please be more specific?

We added the following statement to address that “OMVs can have numerous functions in the bacterial life style such as to deliver their signaling molecules into a host, serve as weapons in the inter-species environmental competition and the release of misfolded proteins [20]

Line 55: “The composition and content of OMV have been investigated” Authors then go to comment on the lack of in vitro studies addressing the effect of such composition on the host without mentioning what components are or whether there are in vivo studies reported in the literature.

We changed that sentence and added another statement to further explain it “The composition and contents of B. burgdorferi OMVs have been investigated, however, knowledge of their ex vivo effects on host cells is still limited [18, 21]. Several biopsy and microscopic examination studies have shown OMVs are present within human tissues and near host cells; and they have also been observed at distance sites from the bacterial infection site [19]….”

Line 66: The sentence in short reads “while immune responses are recognized to mediate the reaction to foreign antigens much less is known about the ability of OMV to alter homeostasis in neurons”. This needs to be rewritten in a way that shows a logical flow of information between cell responses, neurons, and OMV.

We modified that statement to the following to hopefully better explain our point of view in there “Innate immunity is known to detect foreign antigen molecules and triggering signaling pathways that result in rapid secretion of myriad of inflammatory cytokines and chemokines. However, there is little known about the immune response to OMVs and whether they could act as key elements on certain host cells such as neurons and their homeostasis to compensate for environmental changes.”

Methods:

Line 312: Please specify that it is Borrelia burgdorferi the pathogen used in the infections in here too or move/condense with line 322.

 We added the suggested “B. burgdorferi” before infection

Line 319: B. burgdorferi can be mentioned once in this sentence

We removed the 2nd “B. burgdoreferi” and it now reads as “To collect B. burgdorferi bacteria for experiments, they were centrifuged at….

Line 329: Infection timepoints should be described in the method section too.

We added the infections timepoints

Line 381: Ultracentrifugation should be renamed "Isolation of Outer membrane vesicles (OMVs)" or something similar and moved after the description of the cell culture methods.

We renamed the “Ultracentrifugation” to "Isolation of Outer membrane vesicles (OMVs)" and moved that up after the cell culture methods section. 

Results:

It is not clear why internalization in HEK293 cells was made. Especially, when/if part of the reason using this cell line here was to compare reductions in superoxide with the neuroblastoma cells and the data is not shown. I suggest including this data as a comparison if an important claim of this study is that “internalized extracellular factors might be responsible for the rapid superoxide reduction”

Although we had primarily used BE2C cells in this study but we have been back and forth on including or excluding some data we had using the HEK293 but we decided to include the internalization. So that data/figure is added into the revised manuscript and we also included that B. burgdorferi internalization can be observed at 24 hours in HEK293 as well (Figure 2, A). 

Figure 4: unpaired Mann-Whitney test (compare two groups) was used to determine significance between the untreated cells and each time point? This is important because the description of the statistical analysis only mentioned the Kruskal-Wallis test (compare 3 or more groups). I suggest describing in the text (not only in the figure but also in the result section) each test performed to make the reader understand more easily the significance of every comparison. For instance:

“Amplification curve analysis showed that BE2C cell SOD2 expression for infected cells was significantly lower at all infection time-points in comparison to untreated cells (XXX test, p=0.005)”

Line 122: “Quantitative densitometry analysis results showed that BE2C cell supernatants had increased levels of three of several chemokines

We initially had done comparison of more than two groups together, so for those we had used the Kruskal-Wallis test. However, since in this study we only report and show data where the comparison was only done between two groups we removed the Kruskal-Wallis in methods, figure legends and results sections and added the Mann-Whitney test only.

Reviewer 2 Report

The manuscript “Effect of Borrelia burgdorferi Outer Membrane Vesicles on Host Oxidative Stress Response” by Keith Wawrzeniak, Gauri Gaur, Eva Sapi and Alireza G. Senejani” is well written and easy to follow.

A few issues that will enhance the quality of the manuscript.

  1. B. burgdorferi induced cytokine secretion by BE2C cells after 24-hour co-culture was evaluated using an antibody panel and quantitated by densitometric scanning. It would be more appropriate to use standard ELISA assays to quantitate the cytokines that were identified by the antibody array.
  2. Correct spelling and grammar errors

Author Response

Author's Reply to the Review Report (Reviewer 2)

We thank the reviewer for the valuable comment on our manuscript. We have used the Human Cytokine Array C5, which has a Sandwich ELISA specificity feature. This allowed us to monitor and detects 80 Human Cytokines. In our future studies we plan on further investigating some of those cytokines plus a few other candidates recommended by colleagues in more detail. For that we would use some other means such as Elispot, Western blot or individual standard ELISA assay to quantify them. These experiments will likely take the better part of a year and we feel they are outside the scope of the present study. We have made further correction on spelling and grammar in the latest submission

Round 2

Reviewer 1 Report

The authors had addressed the comments.

Author Response

  • We thanks the reviewer to review our manuscript.